# Food Credence Attributes: A Conceptual Framework of Supply Chain Stakeholders, Their Motives, and Mechanisms to Address Information Asymmetry

**DOI:** 10.3390/foods12030538

**Published:** 2023-01-25

**Authors:** Peggy Schrobback, Airong Zhang, Barton Loechel, Katie Ricketts, Aaron Ingham

**Affiliations:** 1Agriculture and Food, Commonwealth Scientific and Industrial Research Organisation (CSIRO), Brisbane, QLD 4067, Australia; 2Health and Biosecurity, Commonwealth Scientific and Industrial Research Organisation (CSIRO), Brisbane, QLD 4102, Australia; 3Environment, Commonwealth Scientific and Industrial Research Organisation (CSIRO), Brisbane, QLD 4102, Australia; 4Agriculture and Food, Commonwealth Scientific and Industrial Research Organisation (CSIRO), Canberra, ACT 2601, Australia

**Keywords:** assurance, asymmetric information, certification, credence, food, attributes, demand, labels, quality, safety, signals, supply, stakeholders

## Abstract

Food credence attributes (e.g., food safety, organic, and carbon neutral production methods) are quality characteristics of products that cannot be assessed by buyers at the point of sale without additional information (e.g., certification labels). Hence, the ability to access credence attributes of a particular product can result in a situation termed as asymmetric distributed information among supply chain stakeholders (e.g., producers, processors, wholesalers, retailers, consumer) where one party of a market transaction is in possession of more information about a product than the other party. This situation can lead to potential inefficiencies, e.g., misinformation, risk of food borne illness, or opportunistic behavior such as fraud. The present study sought to develop a conceptual framework that describes a) the motivation for key stakeholders to participate in the market for food credence attributes, b) the type of food credence attributes that key stakeholders provide, and c) current mechanisms to address the issue of information asymmetry among the stakeholders in the food system. The study was conducted using an integrative literature review. The developed framework consists of two components: a) the food supply chain and b) the attribute assurance system among which multiple links exist. The findings suggest that retailers, processors, NGOs, and government authorities are influential stakeholders within the supply chain of food credence attributes by imposing food quality standards which can address information asymmetry among food actors. While the credence attribute assurance system (e.g., food standards, third party food attribute assurance providers) can potentially address the issue of asymmetric information among market stakeholders, a range of issues remain. These include food standards as a potential market entry barrier for food producers and distributors, limited food standard harmonization, and communication challenges of food attribute assurance (e.g., consumers’ signal processing, signal use and trust). The syntheses presented in this study contributes to stakeholders’ (e.g., supply chain actors, scientists, policy makers) improved understanding about the components of the credence food system and their integration as well as the drivers for change in this system.

## 1. Introduction

Food products can be considered as bundles of multiple quality attributes (e.g., appearance, taste, price, organic production, food safety, origin, nutritional value). In deciding which product to purchase, buyers within the food market (e.g., wholesalers, retailers, consumers) objectively or subjectively assess and compare food quality attributes against one another depending on available and trusted information [1,2,3,4,5].

The theory of information economics suggests that food quality attributes can be categorized into search, experience, and credence attributes. Search attributes describe food characteristics that can be verified by buyers (e.g., importers, retailers, consumers) prior to purchase through direct search/inspection (e.g., visual or physical assessments) [6,7,8]. Examples for search attributes include appearance, color, price, and brand labels. Experience attributes are food characteristics that can be verified by buyers only after consumption (i.e., experience) of the product [6], e.g., sensory attributes such as taste and texture.

Credence attributes are food quality characteristics that cannot be judged or assessed independently by buyers at the time of sale through search (e.g., price, appearance) or experience (e.g., taste, texture) [9] without additional, often costly information [10,11]. Examples include environmental sustainability of production/distribution processes (e.g., carbon neutral, eco-friendly, organic [12], animal welfare [13], and food safety [14] (see more examples provided in Table A1 in the Appendix A)). Buyer’s inability to assess the presence of credence attributes at the point of sale without additional information (e.g., labels) implies that one party in the market transaction is in possession of more information (e.g., producer) than the other (e.g., retailer, consumer). This situation is known as information asymmetry within the food market or among supply chain stakeholders [15,16].

The presence of asymmetric information among market actors such as supply chain stakeholders (e.g., producer, processor, wholesaler, retailer, consumer) offers ground for market inefficiencies that can lead to market failure (e.g., free-riding, moral hazard, and adverse selection) [17,18]. For example, the benefits that buyers believe they pay for (e.g., organically produced food products) may in fact not be provided by the seller as claimed. This situation is known as moral hazard on the supplier side, i.e., opportunistic behavior such as fraud or the provision of misleading information [11,17]. Furthermore, the existence of asymmetric information about food credence attributes can expose buyers to risks [18,19]. For example, there are individual health risks for the credence attribute of food safety, where unsafe food consumption due to inadequate information/communication, can become a risk to the public (health) good in the case of a foodborne disease outbreak [19,20,21].

Hence, from an economic perspective, asymmetrically distributed information about food credence attributes and associated risks can cause a failure of the competitive food market [18,22]. This implies that the existence of a market for food credence attributes depends on the level of information provided to buyers and on buyers’ trust in use of such information [10,18].

The literature offers a large range of studies which empirically assess consumer demand for food credence attributes [23,24,25,26] and selected aspects relating to credence attribute assurance such as food standards [21,27,28], food certification [29,30], and food labelling/signaling [16,31,32]. However, missing in the literature is a clear overview that synthesizes a) the motivation for key stakeholders to participate in the market for food credence attributes, b) the type of food credence attributes that key stakeholders provide, and c) current mechanisms to address the issue of information asymmetry among the stakeholders in the food system.

This study aimed to address this gap by developing a conceptual framework that summarizes the roles and incentives of key stakeholders in the market for food credence attributes and current mechanisms to address the issue of information asymmetry among the stakeholders in the food system. Specific research questions included the following: What are the components of this conceptual framework? How do these components link with each other? Can information asymmetry be completely resolved using currently available mechanisms?

The study was conducted using a literature review and takes a supply chain system perspective on agri-food products as a proxy market for foods with credence attributes (i.e., where a market exchange between various supply chain stakeholders occurs).

The findings from this study offer a theoretical contribution which may improve food supply chain stakeholders’ (e.g., supply chain actors, scientists, policy makers) understanding about the components of the credence food system and their integration as well as the drivers for change in this system.

## 2. Methods

To develop a conceptual framework that summarizes the roles and incentives of key stakeholders in the market for food credence attributes and current mechanisms to address the issue of information asymmetry, an integrative literature review [33,34,35] was conducted. The literature was assessed in a manner that contributed to a) identifying potential components of the proposed framework; b) mapping and describing these components and linkages between them; and c) synthesizing the findings. Importantly, the scope of the review was not to identify the state of knowledge in this research area and appraise the existing literature in the context of food credence attributes (e.g., systematic review which typically focusses on a relatively narrow topic). The aim of the review was rather to identify and describe key themes in the existing literature that can be used to define components of the market for food credence attributes and their integration.

The procedure used to search, assess, and synthesize the literature is illustrated in Figure 1. Google Scholar and Web of Science databases were employed to search for relevant publications which included journal articles and the grey literature (e.g., reports). The publication search was not restricted to a specific time period and only included records in English language. Terms included for the initial keyword and title search were combinations of: “food”, “credence”, “attributes”, “stakeholders”, “assurance”, “certification”, “verification”, “system”, “demand”, “supply”, and “market”. These terms were combined by the AND (e.g., for food, credence, attribute) or OR (e.g., for assurance, certifying, verification) Boolean operators.

The inclusion process was based on an initial assessment of publication titles and abstracts that related to the research topic. Duplications and records of low quality (e.g., some reports, topic related publications with limited research content value) were excluded in the initial assessment stage of publications (e.g., 127 records). A total of 243 publications were identified for inclusion in the qualitative analysis. All publications that passed the inclusion criteria were assessed in full, and their content was categorized by identified themes, e.g., supply chains stakeholders, food standards, and third-party certification. The qualitative analysis of the literature review included theme refinement (e.g., identification of the main themes and sub-themes), theme description (e.g., narrative describing a theme), and theme integration (e.g., identification of linkages among themes). The analysis and mapping of the identified elements were the building blocks for the framework conceptualization.

In case there were unclear matters identified during the detailed assessment of records or the qualitative synthesis of the content, a more specific literature search using refined terms that describe the respective issue, theme, or sub-theme (e.g., “food”, “wholesaler”, “credence”, “safety”, combined by the AND Boolean operator) were used and the process started over. This iterative process (e.g., building, refining, improving) continued until no additional information on the specific themes and links among themes could be identified, all unclear matters in the design of the conceptual framework were resolved and the research questions of this study (see above) where answered. The synthesis in form of a visual model was developed to portray the identified themes and theme integration.

## 3. Results

A conceptual framework of stakeholders and mechanisms to address information asymmetry in the market for food credence attributes was developed based on the literature review (see Figure 2). The framework includes two key thematic parts. They are a) the food supply chain component; and b) the food attribute assurance component. The first component identifies the issue and scale of information asymmetry about credence foods among stakeholders in the supply chain system, while the second component provides potential solutions to address this issue. For example, within the framework’s supply chain component, information asymmetry about credence attributes typically exists between the upstream and downstream supply chain stakeholders (e.g., indicated by a dashed arrow leading from producer to consumers (Figure 2, marked by (1)). The attribute assurance system component in the framework comprises several aspects that aim to provide solutions to the issues of information asymmetry among supply chain stakeholders, i.e., facilitate information symmetry and buyers’ trust in product attributes (Figure 2, marked by (2)). Essentially, if information symmetry can be achieved, credence attributes will be transformed into search attributes (Figure 2, marked by (3)) [36,37]. This framework can be applied to national and global food markets or supply chains.

The following sections describe the two components of the framework (e.g., key themes, sub-themes) and links between them. While a large literature exists for some elements of the framework such as food standards, food certification, and communication of food quality attributes, they are only briefly outlined here in the context of this study. Interested readers may refer to the cited literature for more details on these elements.

### 3.1. Food Supply Chain Component

The food supply chain component in the framework comprises key stakeholder groups. These stakeholders include food producers (who are here bundled together with production input providers for simplicity, e.g., nurseries, seed providers), logistics providers, processors, wholesalers, retailers, and consumers (Figure 2). These key stakeholders are directly involved in the physical exchange and distribution of foods with credence attributes and may therefore be directly exposed to potential information asymmetries within the supply chain (indicated by dashed arrow in Figure 2). Other stakeholders may be indirectly involved in the physical distribution of credence foods, such as investors in agricultural production, investors in food retail or research and development organization. These indirect stakeholders are not further considered here for simplicity. The framework illustrates that the physical distribution of the food product and the relevant information about quality attributes among supply chain actors may not occur simultaneously.

In this section, we identify the role and motives of key supply chain stakeholder groups in participating in the market of food credence attributes. The type of credence attributes which these stakeholders provide will also be identified. Key findings are summarized in Table 1. We outline stakeholder roles in creating and overcoming asymmetric information issues and highlight the potential pressures that may affect their participation. The framework assumes a long agri-food supply chain (e.g., multiple intermediaries between producers and consumers), which reflects modern food networks in high-income countries as well as export supply chains [38]. The framework also considers limited vertical coordination (e.g., information sharing) between supply chain stakeholders as a simplification to illustrate the potential scale of the asymmetric information issue.

#### 3.1.1. Food Producers

The primary role of producers within the food supply chain (Figure 2) is to produce food products including food credence attributes (e.g., environmental sustainability of production, social/ethical responsibility of practices, animal health and wellbeing during production process, additive free production, see Table 1 and Table A1 in the Appendix A) and ultimately supply these to buyers (e.g., processors, importers, retailers, consumers) who may also demand these food quality attributes.

From a business perspective, the producer’s incentive to participate in the food market as supplier is to make a living, to maximize profits from the production of the food products and subsequently to ensure enduring economic viability of their business [39,40,41] (these are hereafter referred to as *business incentives*). The supply of food quality, specifically the safety and biosecurity of food for human consumption, is a key characteristic to gain market access [41,42,43].

Hence, food producers will supply food with specific quality attributes including information about these attributes (e.g., certification of attribute claims), if there is sufficient and continuous buyer demand, if the supply is profitable for producers (e.g., compliance with private food standards), or if they are required to do so (e.g., compliance with food regulation in domestic and export markets; hereafter referred to as *regulatory incentives*) [10,11,16,28].

Profitability implies that the marginal costs of producing foods with credence attributes (e.g., growing the food product, obtaining attribute certification, and other transaction costs) are less than the marginal benefits from the sale of the product (e.g., product price at farm gate). A potential premium price paid by buyers may be a key incentive for producers to provide credence attributes [43]. Contributions to maximizing business profits from the supply of credence attributes can stem from product differentiation (i.e., conventional product vs. organic product) and associated gains in market share (e.g., increased sales) or from avoidance of costly events such as foodborne disease outbreak with associated liabilities [16]. Hence, a prerequisite for producers to supply food credence attributes is having sufficient understanding about buyers’ demand, including their quality requirements for food (e.g., specific attributes demanded, willingness to pay for attributes, trust in attribute claims verification mechanisms) [11,44,45].

Other indirect economic motivations for food producers to supply food credence attributes, specifically environmental and ethical/social focused food quality attributes, are social pressures from stakeholders such as the local community, consumers (e.g., ‘pull’ for certain production methods), government (e.g., best-practice guidelines for improved animal welfare production methods), investors, and the media [44,46,47,48,49]. Drivers for increasing social pressures from these food system stakeholders on food producers include, for example, growing consumer concerns which are amplified through increased awareness/knowledge (e.g., through advanced communication technology, media-savvy population), urbanization, incomes, education, and associated changing social norms [49,50,51]. Producers’ economic motivation to respond to these pressures addresses their objective to maintain their social license to operate (i.e., society provides producers the approval to conduct current food production activities) [49,52,53].

Other non-economic factors that can motivate producers to supply credence attributes come from their own attitudes about the environment, demographics, health concerns, lifestyles, and ideology which can also influence the ethos of their entire business model [11,54,55,56].

#### 3.1.2. Logistics Providers

Logistics providers physically transfer food products from one location to the next within the supply chain (see Figure 2, arrows linking downstream and upstream supply chain stakeholders). Logistics providers typically contribute to the supply of credence attributes such as food safety and product traceability (e.g., manual or more sophisticated systems) (Table 1) [57,58]. Business incentives (see above) are key motivations for these supply chain actors to participate in the market for credence attributes. Logistics provider practices are commonly mandated by national food and animal welfare regulations (i.e., public standards for transportation of live animals [59]) to ensure that these food quality attributes (including their production) are maintained with the product in their care, notably to minimize human health risks and animal welfare risks [57,58].

#### 3.1.3. Food Processors

The role of food processors in the food supply chain is to consolidate the raw product from multiple producers, to clean and grade it [60]. Their role can also include the deliberate change or transformation of the raw food product (e.g., in some cases inedible products such as live animals) into shelf-stable and palatable foods (e.g., meat cuts) for human consumption [60,61]. Packaging of the processed food products is another added value that processors offer [60].

Food processors can also supply credence attributes such as religious/cultural animal slaughter methods [62,63], animal welfare at slaughter [64,65], food safety [66,67], product traceability [68], and social/ethical responsibility (e.g., labor rights) [69] (Table 1).

Business and regulatory incentives (see above) are also key motivations for food processors to participate in this credence food market [68]. One such incentive is the opportunity for product differentiation, e.g., religious animal slaughter methods. Likewise, food processors are typically required to comply with national food regulations (e.g., public food standards on hygienic processing and transportation of food products for human consumption) in avoidance of risky and costly events (e.g., outbreak of food-borne illness due to unsafe food handling). To achieve access to premium and specialized markets, food processors are also involved in self-regulating their operations in terms of setting quality standards in their food procurement (e.g., private standards for size, appearance) and can opt to adopt standards for specific processing methods (e.g., religious animal slaughter methods).

Exemplifying a mix of business and regulatory incentives, meat processors increasingly experience pressures from society and consumers to offer credence attributes such as animal welfare at slaughterhouses which include humane handling, stunning and slaughter of animals e.g. [70,71]. National best-practice guidelines, government legislation e.g. [72], and intergovernmental organization standards e.g. [73] have set incentives for processors to improve their animal welfare performance. It should be noted that the literature offers limited examples for indirect economic pressures on processors that handle non-meat food products such as horticulture and seafood processors e.g. [74]. This leads to the assumption that the societal pressure on non-meat food processors and, subsequently, consumers’ expectations about credence attributes such as food safety provided by these supply chain actors, may be less than for meat processors or food producers (see above).

#### 3.1.4. Food Wholesalers

The role of food wholesalers (e.g., distributors, brokers, agents) is to consolidate and coordinate the distribution of food products between upstream and downstream supply chain stakeholders (e.g., producers and retailers or processors and retailers, simplified in Figure 2).

Their motivation to participate in the supply and distribution of food products with credence attributes is driven by business and regulatory incentives (see above). The provision of added value such as food safety (e.g., handling of products) [75,76] and product traceability (e.g., record keeping of consignments, information disclosure, vendor traceability; see Table 1) are typically compulsory for food wholesalers as determined by national food regulations [76,77]. Interestingly, no studies could be identified that outline societal pressures on wholesalers with respect to the provision of credence attributes.

#### 3.1.5. Food Retailers

The role of food retailers (e.g., retail stores, catering, hospitality services) is to procure food products in bulk volume from upstream food system actors (i.e., producers, processors, wholesalers) and distribute them to consumers. From a value-adding perspective, retailers provide credence attributes such as food safety and food traceability [78] and social/ethical responsibility (e.g., fair treatment of labor, labor health and safety) [79] (Table 1). However, they may offer value adding services such as advertising and branding of the product, which includes the promotion of product credence attributes [80,81].

Business and regularly incentives (see above) also apply to retailers as motivation for participating in the market for food credence attributes [82]. Retailers’ supply and distribution of food products with credence attributes (e.g., food safety, transparency, organic- or carbon-neutral food) can be considered as a form of product range differentiation to satisfy consumer demands (e.g., product quality) [28,80,81,83,84]. Hence, key motivation for retailers to participate in the market for food credence attributes is to increase consumer satisfaction (or prevention of the loss of market share), to gain and maintain a positive reputation, brand assurance, and to create a competitive market advantage (i.e., attract a specific consumer cohorts, gaining market share through supply of quality food) [28,41,45,81,82,85].

Consumer demand for credence attributes can been seen as a profitable business opportunity for retailers, like other food chain actors described above [65]. However, this requires an understanding of consumers’ demand for food quality attributes. Larger retail chains increasingly use their managerial abilities, direct links to, and knowledge about consumers to develop commercial strategies (e.g., private food standards to control food quality in their supply chains) that address this market differentiation opportunity to develop and maintain a competitive market advantage [21,81]. Retailers’ increasing supply chain governance implies that they strategically respond to consumer demand for food quality through measures that focus on gaining and maintaining consumer trust [86,87]. This is completed by using mechanisms such as private standards (see details in Section 3.2.1) to coordinate and control quality in retailer’s supply chains, and product signals (e.g., price, labels, and information, see details in Section 3.2.4) to address information asymmetry among food chain actors (see Figure 2, more detail is provided in Section 3.2.4) [28,38,86,88]. Retailer’s growing supply chain governance is seen as one of the key drivers for restructure in the production and distribution of agri-food products [87,89]. Retailers also have a major influence in shaping consumer food choices and preferences through pricing and promotion, which is commercially motivated by retail market competition [81]. Therefore, retailers also have a role in influencing consumer food choice [41].

Factors that indirectly contribute to retailers’ economic incentive in supplying products with credence attributes include social and ethical responsibility (i.e., business model and ethos) and subsequently their social license to operate and corporate legitimacy, which can also affect consumer’s retail preference [38,90,91,92].

#### 3.1.6. Consumers

Consumers are the end users of the food products that are supplied and distributed by producers and various other supply chain intermediaries (Figure 2).

From an economic perspective, consumer motivation to participate in the market for food credence attributes include the satisfaction of their basic human needs (i.e., consumption of safe and nutritious food) and the satisfaction of individual wants (e.g., consumption of locally, sustainably, or ethically produced food) [93,94]. As economic actors in the food market, consumers make choices about the allocation of their scarce resources (e.g., weekly budget) to satisfy their needs and wants [95].

Consumers’ objective to satisfy their food needs and wants is associated with their risk perspectives on food quality attributes. These risk perceptions can be divided into a) food risks related to human health (e.g., expected quality attributes such as food safety, nutritional properties) [93,96,97,98] and b) risks related to consumers‘ ideological food requirements (e.g., desired production methods that satisfy their individual wants) [93].

Human health-related food quality attributes (e.g., safety, absence of harmful additives) are necessary and expected by all consumers cohorts, as part of the basic human needs, but are not always provided [99]. Government regulation (e.g., public food standards, see Section 3.2.1) focusing on safe production and distribution of food for human consumption (i.e., public health) typically exists to ensure that these consumer expectations are met [100].

Other credence attributes may only be demanded by specific consumer cohorts, as an added value that meets their individual wants/preferences [93] (Table 1, Table A1). Particular ideological values that are captured in food attributes (e.g., animal welfare) give these specific consumer groups a higher utility compared to conventional food products [65]. These consumers cohorts are typically willing to pay a premium price for the presence of their desired food credence attributes [101,102,103,104].

Consumers objectively (e.g., choice based on verified attributes claims) or subjectively (e.g., choice based on unverified attributes claims) assess food quality attributes prior to purchasing the product [105]. Choice for food quality can be influenced by signals (e.g., labels) that provide information and verification of credence attributes claims [106] (see more details in Section 3.2.4).

Factors that contribute to consumers’ increasing demand or ‘pull’ for food credence attributes include, for example, risk perceptions arising from food safety incidents, socio-economic development (e.g., education, increasing income and subsequent affordability), increased awareness and reflective consumerism (e.g., social and environmental impact of food choices), fashion, product marketing, and the influence of media and advocacy groups [10,38,51,85,107,108,109].

The level of information about the credence attribute supplied to buyers along the supply chain, and associated uncertainties about food product attributes, can affect purchase decisions of supply chain stakeholders [110]. Figure 2 illustrates that a quality assurance mechanism is needed in the market for food credence attributes to address the issue of asymmetric information among the supply chain stakeholders. The following section outlines how quality assurance mechanism can address the issues of information asymmetry.

### 3.2. Attribute Assurance System Component

The role of an attribute assurance system within the conceptual framework is to facilitate information symmetry between supply chain stakeholders and to create buyers’ trust in attribute claims (i.e., trust in presence and authenticity of credence) (Figure 2).

This component of the framework comprises the following: a) private and public food standards; b) supply chain actors’ adoption of food standards; c) verification of supply chain actor compliance with food standards through conformity audits; and d) the communication of attribute assurance via signals (Figure 2) [45,111,112]. This section briefly outlines the elements as well as key stakeholders (e.g., government, food supply chain stakeholders, intergovernmental and non-governmental organizations) and the motivations for their involvement in the governance of food attribute assurance systems.

#### 3.2.1. Food Standards

Food standards are the foundation for food quality attribute assurance systems [21,27,85,113]. Food standards are defined as “documented agreements containing technical specifications or other precise criteria to be used consistently as rules, guidelines or definitions, to ensure that materials, products, processes and services are fit for their purpose” [114]. They are broadly categorized into public and private standards that determine the product performance or composition (e.g., absence of additives, non-genetically modified foods) and how food is produced (e.g., organic, carbon neutral), processed (e.g., halal, kosher), handled, and distributed (e.g., food safety consideration) [21,27,45,85,115].

##### Public Food Standards

Public food quality standards are based on public law (e.g., regulation/policies) which can be set by national, regional (e.g., European Food Safety Authority) or intergovernmental organizations (e.g., FAO/WHO Food Standards Program) [21,27,116,117,118,119].

Public standards typically have a social welfare objective, such as protecting public goods (e.g., population health and food safety) and common goods (e.g., environmental assets) that considers interests of all actors in the food system (e.g., producers, distributors, and consumers) [20,21,27,116]. Public food standards can influence the supply (e.g., composition standards) and demand (e.g., food labelling standards) for food products. They can also focus on ‘fair trade’ practices (i.e., arrangements to ensure that food is produced in an equitable manner) that link to food quality attributes [116,120,121]. This type of food standard can either be compulsory or voluntary in its conformity requirements [122]. Examples of public standards include national food storing, handling, and disposal requirements, requirements for health and hygiene of food handlers, cleaning and sanitizing of specific equipment, labelling [123], as well as the Codex Alimentarius which offers international standards, guidelines, and codes for safety, quality, and fairness in food trade [124].

Hence, the role of government authorities within the market for food credence attributes is to set public regulations (e.g., minimum benchmarks for food safety and health, labelling requirements for ingredients and origin) that facilitate safe, affordable, and nutritious food consumption [20,27,38,43,85,125,126]. However, budgetary constraints typically limit regulatory activities of government authorities within the market for food attributes beyond ensuing that public goods are not at risk, which offers opportunities for private standards [85].

##### Private Food Standards

Private standards for credence (and non-credence) foods are set by stakeholders that operate along the food supply chain, such as large retailers, processors, industry associations (e.g., Meat Standards Australia set by Meat and Livestock Australia [127]) (see Figure 2 link between framework components), or interest-based groups (commonly referred to as non-governmental organizations (NGOs)) [21,27,28,128]. This form of food standard is frequently seen as ‘going beyond’ the requirements of public standards [85,129] and typically operate alongside them [130].

Private food standards set by supply chain actors are considered as business-to-business (B2B) (e.g., retailer efforts to control supply quality, vertical coordination of supply chain) or business-to-consumer (B2C) (e.g., certification labels as a quality assurance signal for consumers) quality assurance benchmarks [27,30,85].

Private food standards represent the interest of the stakeholders who develop and promote the standard. Hence, their objectives can vary according to their pursued aims (e.g., for retailers this typically includes profit maximization and/or standardizing product and distribution requirements across suppliers for quality control; for NGOs it typically includes work toward global harmonization of standards and to promote the interests of society) [21,27,130]. Private food standards are voluntary from a legal adoption perspective although there may be a compulsory requirement for access to the specific associated supply chain or market [21,85,87,122]. This type of food standards is considered as a form of supply chain governance or coordination as well as a means of competitive positioning in markets for food products (e.g., product differentiation by setting higher quality standards) [130,131,132]. Private food standards typically also reflect the interests of buyers and their adoption are generally mirrored in the product price [21,85,130]. Developing private food standards can be costly for standard owners since it includes the establishment of criteria/benchmarks, standard accreditation, and establishment of monitoring and enforcement procedures [30,132]. Examples of private food standards include norms, that retail groups set to control food quality within their procurement, and collective international standards such as Global Good Agricultural Practices (GlobalGAP) [28,133].

Henson and Reardon [130] and Fulponi [85,130] argued that the drivers for the increasing role of private standards within the food markets are the increasing legal liability of food suppliers, consumer’s concerns about product safety and quality, and associated reputational and/or commercial risks if consumer demands are not met. Other authors claim that private food standards are becoming predominant drivers for the supply of higher quality foods and for the broadening scope of food quality (e.g., promotion of emerging credence attributes) [e.g., 28, 130].

##### Standard Harmonization

The high number of food standards globally, specifically standards targeting food safety, and their varying criteria (e.g., differences in food safety regulations and private standards for organic production and between nations) have been identified as a market access barrier for food suppliers (e.g., a potential form of protectionism, non-tariff technical measures) [120,134,135]. Limited food standard harmonization (e.g., lack of uniform norms) has implications for producers. For example, small-scale producers and producers from low-income countries aiming to participate in international food trade may financially not be able to adapt their production methods to different but similar food standards that food importers set (e.g., due to high costs and time involved in adoption and compliance) [120,134]. Limited food standard harmonization can also lead to a reduction in food trade volume, which can affect food security [120,134]. Food standards have also been identified as catalysts for quality and technology upgrading in the food industry [112,136].

Driven by lowering barriers for international trade in food, global harmonization of public food standards has advanced (e.g., Codex Alimentarius, World Trade Organization’s agreement on the application of sanitary and phytosanitary standards (SPS)) [137,138]. However, harmonization of private standards (e.g., various standards for one credence attribute) proves to be more challenging given the varying motivations of private supply chain stakeholders in imposing these (e.g., product differentiation, gaining commercial advantages) [135,139].

#### 3.2.2. Adoption of Food Standards

Standards provide the foundation for the food attribute assurance system (Figure 2, link between framework components) and producers, processors, wholesalers, and retailers may variously be required, or choose to, adopt these standards to gain access to the associated market of food credence attributes [43,115].

Different degrees of freedom exist in the adoption of food standards. For example, at one extreme, compulsory public standards are imposed (e.g., for food safety) by government authorities that can act with coercive power to ensure compliance with regulations and policies [21,27,129]. Non-adoption of compulsory public standards may, depending on public monitoring and enforcement, incur economic losses due to fines and temporary or permanent cessation of operations, loss of rights of market access, and loss of reputation [43]. Such commercial risks give food suppliers and distributors incentives to ensure compliance with compulsory public food standards [43]. However, some public food guidelines and codes can also be voluntary in their adoption, e.g., guidelines for *Hazard Analysis Critical Control Points* (HACCP) procedures [21,27,140]. The other extreme on the freedom of adoption scale are private voluntary standards [27,129]. The adoption of private food standards is choice-based for suppliers and distributors, meaning these stakeholders can opt to adopt these standards if they wish to supply their product to buyers who impose these. However, if private standards are widely imposed by buyers and are a requirement to access specific supply chains, the standards become de facto compulsory [27,140]. The adoption of food standards by food suppliers is typically associated with costs (e.g., transaction and implementation costs), which, if too high, can be an access barrier for suppliers to participate in the credence food markets [87].

#### 3.2.3. Food Standard Conformity Audits

Government authorities and buyers of credence foods may require proof or assurance that specific food standards they imposed are met by suppliers and distributors. This is a central part in overcoming information asymmetry among the supply chain stakeholders and can be addressed by conformity audits.

Conformity audits are assessments (e.g., inspections of documents, test of processes and products, check of facilities and practices) which verify that a product, process, or person conforms with the requirements set by a food standard [27,45,111]. The certificate issued by a certification body is intended to confirm and guarantee compliance with the requirements of a food standard [28,111,132].

There are different types of certification bodies that can undertake conformity audits and issue conformity certificates [27,111]. These include first parties (e.g., self-declaration of food quality claims, producers who issue quality guarantees), second parties (e.g., buyers through contracts that specify quality requirements to be fulfilled, retail brands), and third parties (e.g., independent/neutral organizations that are not directly involved in the exchange of the food product) [30,111,131]. Third parties may be categorized as trusted and independent actors that serve as intermediaries between parties in market transactions (e.g., sellers and buyers) [30,131,132]. However, the operational independence of third-party certification providers may vary, since in many cases food suppliers pay them to conduct certification audits, whereas others are indirectly funded through industry production levies or by the government, which may affect buyers’ judgement about the credibility of the conducted audits [141].

Third-party certification bodies can be private (e.g., commercial firms, NGOs) or government organizations [111]. In some markets public food attribute verification schemes are dominant, while in other markets private verification schemes prevail [30]. Private third-party certification bodies typically require formal recognition (e.g., attestation of competency to carry out food certification audits) by an accreditation body [30,45,111,142]. Accreditation is a mechanism to standardize and regulate third party certification bodies (i.e., to minimize fraud and rent-seeking behavior) in the broader food market [30,45,131]. Accreditation of food certification bodies is commonly performed by a national accreditation authority (i.e., government), NGOs, or industry organization [45,111,131].

Importantly, conformity audits of food products, processes, premises, and persons are associated with costs for the stakeholders (e.g., producers, processors) who are required to provide verification of compliance with the food standards imposed by buyers [11,27]. These stakeholders will assess the marginal costs and benefits of meeting food standards and obtaining certification. This cost-benefit assessment subsequently leads to a decision about the commercial value of overcoming these market entry barriers or not [11,30,122,129] (see Section 3.1.1). On the other side of the ledger, buyers (e.g., retailers, consumers) may demand assurance of compliance with the food standards they impose (e.g., third-party certification) as a risk management strategy (e.g., reputation, liability) and to reduce their own conformity assessment or monitoring costs (e.g., control over suppliers) [27] (see Section 3.1.5).

#### 3.2.4. Communication of Attribute Conformity with Standards via Signals

Suppliers (e.g., producers, processors, retailers) of credence food attributes may choose to communicate the presence of these quality characteristics to their buyers. Various means serve as signals or cues of the quality of unobservable food quality attributes, including quality labels (e.g., logos, marks) placed on product packaging through product and corporate brands, producer or retailer guarantees (i.e., self-declaration labels), third-party certification, and suppliers’ personality and internet presence [18,21,32,36,80,143]. Such signals for food quality (e.g., brand label, third party certification mark) can reduce buyers’ perceived risks associated with information asymmetry, and can guide their decision making [141]. Caswell [36] argued that information such as food quality labels (e.g., certification mark, retailer guarantees) offer buyers signals that can transform credence attributes into quasi-search attributes. Among these communication tools, third-party certification logos can lead to higher buyer trust in food quality attributes compared to other product quality-signaling tools [141,143,144,145].

However, credence attribute signals are only effective in addressing information asymmetry if buyer preferences are met (e.g., individual quality attributes vs. bundles of quality attributes), the information is processed (e.g., cognitive ability to deal with provided signals, familiarity with certification labels and associated food standards), and used (e.g., trigger product-purchase decision) by the targeted audience [21,80]. For example, information overload (e.g., amount of information on the product), consumer confusion and knowledge (e.g., ability to differentiate between many certification labels/food standards, vague and varying criteria that certification labels represent, buyer familiarity with credence attributes), buyers expectations (e.g., sensory expectations of credence foods), and their potential indifference (e.g., due to insufficient credibility of signals) can reduce the efficacy of food quality signals [18,21,37,80,141,144,146,147]. Moreover, buyers’ decision to purchase food products with credence attributes also depends on their trust in, and the credibility of, the attribute assurance system, including standards and certification organizations, as well as their willingness to pay for these attributes [18,32,43]. The challenges involved in effectively communicating product quality information of food products has attracted a large literature [32,37,80,144,146,148,149].

## 4. Discussion

This study aimed to develop a conceptual framework of stakeholders’ motivation to participate in the market for food credence attributes and mechanisms to address information asymmetry in the market for food credence attributes. The synthesis of the literature suggests that there are two key components of the framework: the food supply chain in which information asymmetry among stakeholders exists; and the attribute assurance system which includes the mechanisms (e.g., food standards, food standard conformity audits, communication tools) to facilitate information symmetry and trust in food attributes among supply chain stakeholders (Figure 2).

The findings from the review highlight that food actors such as retailers, processors as well as NGOs and government authorities who impose food quality standards are influential stakeholders in the market for credence attributes. Food producers’ role in this market seems to be less influential as they appear mostly to respond to the demands of buyers, including the supply of food quality assurance, as well as to public food regulations. Logistics providers and wholesalers hold a distributing role in the supply chain of credence foods, for which minimum food safety regulations apply. Food quality assurance providers play a supporting role in this market.

As highlighted in the developed framework, government authorities’ involvement in the market for food credence attributes aims to minimize risks to public and common goods by imposing public standards, which are interventions that facilitate and control the operation of the market for food credence attributes [45]. Government authorities can also act as a mediator between parties in an exchange through provision of certification and accreditation services. Given current pressures on the global food system from environmental (e.g., climate change, diseases, pests, soil and water salinization) and population pressures (e.g., increased demand on food and water supplies, urban encroachment on high quality agricultural land), it is likely that the involvement of governments in the market for credence attributes will increase in future in its function to safeguard public and common goods (e.g., additional public standards and increasing scope of public standards).

The synthesis of the literature also suggests that all actors involved in the supply and distribution of credence foods may ultimately lose their economic incentive to participate in this market if consumers do not demand the products they are seeking to sell (e.g., due to lack of demand for specific quality attributes, insufficient signals that communicate the product quality) [28,44,81], or if regulation does not require the supply of credence foods (e.g., food safety). This implies that consumers are the key actors (the ‘pull’) in the market for food credence attributes (e.g., buyer-driven market) [44,150]. This finding is supported by the rationale for addressing the information asymmetry issue among the food supply chain stakeholders, which includes consumers’ needs for food quality information in their decision making (Figure 2) [18].

The framework demonstrates that although mechanisms exist to potentially address information asymmetry among supply chain actors (e.g., food standards, attribute assurance audits and signals), the transformation of credence attributes into quasi-search attributes is complex [143], specifically the effective signaling of credence qualities to consumers remains challenging [147]. Emerging technologies, such as whole supply chain traceability systems, may offer new communication channels among stakeholders. However, the issues of actor’s awareness and trust in food standards and attribute authenticity signals as wells as actors’ capability to process and use these signals will likely remain challenges, which emerging technologies may not be able to overcome straightforwardly.

A further challenge identified by the framework is the proliferation of food standards worldwide and the need for further harmonization of them. While food standard harmonization would benefit buyers (e.g., increased transparency and buyer confidence in attributes), the literature shows that such harmonization is difficult to achieve due to stakeholders’ vested interests as well as cultural, political, economic, and scientific contexts within which food standards are set [120,151]. However, increased focus should be directed on solving this issue, as it can contribute to buyer’s increasing confusion and subsequent ignorance of attribute assurance signals such as food standards certificates/labels.

While the market for most credence foods (except food safety and biosecurity) is currently relatively small in most countries [152,153], consumer demand for these quality characteristics may grow in future and subsequently increase their market share. Consumer’s future demand for food quality (e.g., sustainable and ethical production methods) may be affected by influences such as awareness, education, and the media, which may be the drivers that will determine the scale of this market. Other supply chain stakeholders such as large retailers may also be able to influence consumer demand for credence attributes in future (e.g., promotion of specific attributes) motivated by their economic incentives (e.g., gaining market share, profit maximization, license to operate) and private standards [28].

However, affordability of credence foods (e.g., most are sold at premium price, except for safety), specifically under current pressures that affect global food supply (e.g., violent conflicts, political tensions, food market speculation, COVID-19 pandemic, extreme weather events due to climate change) and subsequently basic food prices [154], may also mean that only high-income consumer cohorts will be able to afford these products in future. The affordability of credence food products also raises equity concerns with respect to the relative ability of high- and low-income consumers to purchase credence foods. Furthermore, other aspects such as convenience, availability, and time needed to source desired credence foods, can also affect consumer demand and hence the growth of this market [141]. Hence, if suppliers aim to remain competitive in the market for food with credence attributes, they will need to meet consumer’s evolving food quality demands.

## 5. Conclusions

The central issue for food with credence attributes is to overcome the information asymmetry among actors in the market. The developed conceptual framework describes key stakeholders involved in the market, their motives for participation, and the mechanisms to address the information asymmetry. A key finding from this framework is that although mechanisms exist to address the information asymmetry, a range of challenges remain, e.g., need for food standard harmonization, and actor’s trust in and use of quality assurance signals. These challenges may affect the future growth of the credence food market. Therefore, further effort should be put into addressing these matters through targeted research and tailored policies.

Potential scope for further research includes the categorization of challenges in harmonizing food standards and the identification of potential solutions to the increasing proliferation of food standards, e.g., strategies to overcome individual interests of standards owners. Furthermore, continuous research about consumers’ evolving food quality demands, e.g., emerging credence attributes, and consumer’s willingness to pay for these attributes, is crucial for suppliers to remain competitive in the credence food market. Moreover, food suppliers need to continue monitoring consumers’ information needs for purchase decision since addressing these needs is crucial for the credence food market to function effectively and further grow in future.

A limitation of this study is the restricted description of the attribute assurance system components (Section 3.2), which is a result of balancing length of this study and content. Furthermore, while an integrative literature review is an accepted research method [33], it lacks full transparency in the qualitative analysis as only limited guidelines/standards for such assessments exist [34]. This may present a potential risk for a bias in the analysis of the present study.

## Figures and Tables

**Figure 1 foods-12-00538-f001:**
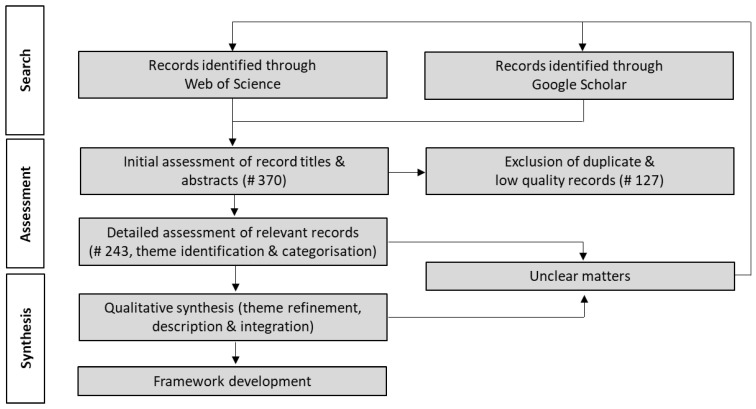
Literature search, assessment, and synthesis procedure. Notes: # refers to the final number of publications considered in the assessment, i.e., after search iterations.

**Figure 2 foods-12-00538-f002:**
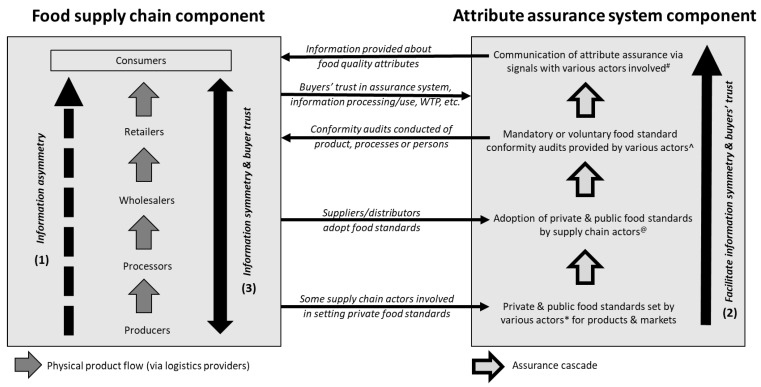
Conceptual framework of stakeholders and mechanisms to address information asymmetry in the market for food credence attributes. Notes: (1)–(3) indicates the sequence of action in addressing information asymmetry among supply chain actors. * Actors involved in private and public standard settings include the government, private supply chain stakeholders, e.g., producer organizations, retailers, non-governmental organizations. @ All supply chain stakeholder groups, except consumers, mandated to adopt food standards (i.e., public standards) or can opt to adopt private standards. ^ Certification providers can include first parties (e.g., producer who issue food quality guarantees), second parties (e.g., retailer guarantees), and third parties (e.g., independent private or governments actors which undertake conformity audits and provide certification). # Communication of credence attributes through signals such as product labels includes producers, processors, retailers, and certification providers. Source: Authors’ summary based on the literature review.

**Table 1 foods-12-00538-t001:** Motives for stakeholders’ participation in the market for food credence attributes.

Supply Chain Stakeholder Group	Motivations for Participation in the Market for Food Credence Attributes	Key Credence Attribute Categories Supplied/Demanded (Examples)
Producers	Maximization of business profits (i.e., product differentiation to gain market share, price premiums), long-run economic viability (e.g., social license to operate), legal requirements (e.g., provision of safe food for human consumption), gaining market/supply chain access, producer’s values/business model and ethos	Supply all attributes as listed in Table A1 (see Appendix A)
Logistics providers	Maximization of business profits (i.e., service provision), long-run economic viability, legal requirements (e.g., safe handling of food for human consumption), gaining market/supply chain access	Supply of food safety and product traceability
Processors	Maximization of business profits (i.e., product differentiation to gain market share, price premiums), long-run economic viability (e.g., social license to operate), legal requirements (e.g., provision of safe food for human consumption), gaining market/supply chain access	Supply of religious/cultural animal slaughter methods, animal welfare at slaughter, food safety, product traceability, and social/ethical responsibility (e.g., labor rights); may demand other credence attributes as part of procurement strategy
Wholesalers	Maximization of business profits, long-run economic viability, legal requirements (e.g., safe handling of food for human consumption), gaining market/supply chain access	Supply of food safety and product traceability, may demand other credence attributes as part of procurement strategy
Retailers	Maximization of business profits (i.e., product differentiation to gain market share, price premiums), long-run economic viability (e.g., social license to operate), legal requirements (e.g., provision of safe food for human consumption), shaping consumer demand, gaining market access, business model and ethos	Supply of food safety, product traceability, and social/ethical responsibility; may demand other credence attributes as part of procurement strategy
Consumers	Satisfaction of basic needs (e.g., food safety) and individual wants (e.g., organic, animal welfare, breed) for food quality attributes	May demand all attributes listed in Table A1 (see Appendix A)

## Data Availability

All data is available from the reviewed/cited literature. No ethical approval was required for this literature review from the author’s institution.

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
