# Peer review of "Food Credence Attributes: A Conceptual Framework of Supply Chain Stakeholders, Their Motives, and Mechanisms to Address Information Asymmetry"

_foods, 2023, doi:10.3390/foods12030538_

Round 1

Reviewer 1 Report

The article is interesting and generally, it deserves to be published with some revisions that are suggested below:

1)     In the Abstract, would you please describe in short your study finding, who are to play a powerful supporting role in this market, e.g., Food quality assurance providers, Food producers’ role in this market seem to be less influential, Logistic providers and wholesalers hold a distributing role in the supply chain of credence foods, and Food quality assurance providers play a supporting role in this market?

2)     In section 4 (Discussion and conclusion), Can you state clearly what the mechanism, how to address the issue of information asymmetry, and what a range of challenges remain for further study

Author Response

Thank you very much for your review. Please find below our responses to the comments. The revised manuscript is attached.

Comments: The article is interesting and generally, it deserves to be published with some revisions that are suggested below:

In the Abstract, would you please describe in short your study finding, who are to play a powerful supporting role in this market, e.g., Food quality assurance providers, Food producers’ role in this market seem to be less influential, Logistic providers and wholesalers hold a distributing role in the supply chain of credence foods, and Food quality assurance providers play a supporting role in this market?

Response: Thank you for this suggestion. The abstract has been revised considering the provided comments (see lines 28-30). Please also note the changed structure of the abstract and a clearer outline of the study’s contribution which was suggested by another reviewer.

Comment: In section 4 (Discussion and conclusion), Can you state clearly what the mechanism, how to address the issue of information asymmetry, and what a range of challenges remain for further study

Response: In the amended manuscript we list the mechanisms used to facilitate information symmetry in lines 601-602. The key issues remaining in addressing information asymmetry include effective communication of attribute presences (discussed in lines 634-643) and food standards proliferation (discussed in lines 644-654). We outline in the conclusion that further research is needed to address these two challenges (see lines 684-692).

Reviewer 2 Report

It is an interesting article in the field of food, to improve some aspects I recommend the following:

Since the article is saturated with citations (which is not bad) and it is a bit complicated to read it fluently, the use of brackets [ ] as established by the journal is recommended.

Some aspects that should be corrected are in the following lines:

Lines 30, 93, 94, 112, 113, 113, 146, 418, 419, and 420: the semicolon (;) is used when separating ideas as in this sentence: a) identifying potential components of the proposed framework; b) map- 112 ping and describing these components and linkages between them; and c) synthesising 113 the findings.

Check-in all of them are separated by commas (,), in some lines even the first clause is missing.

The authors should correct the presentation style in lines 143, 203, 365: there is a period separated from the title number: for example, in section 3 it appears as follows: 3            .Results

Please correct 

Congratulations to the authors

Author Response

Thank you very much for your review. Please find below our responses to the comments. The revised manuscript is attached.

Comment: It is an interesting article in the field of food, to improve some aspects I recommend the following:

Since the article is saturated with citations (which is not bad) and it is a bit complicated to read it fluently, the use of brackets [ ] as established by the journal is recommended.

Response: Thank you for this suggestion. The citations in the manuscript were changed to Vancouver style, including the suggested squared brackets.

Comment: Some aspects that should be corrected are in the following lines:

Lines 30, 93, 94, 112, 113, 113, 146, 418, 419, and 420: the semicolon (;) is used when separating ideas as in this sentence: a) identifying potential components of the proposed framework; b) map- 112 ping and describing these components and linkages between them; and c) synthesising 113 the findings.

Check-in all of them are separated by commas (,), in some lines even the first clause is missing.

Response: Thank you for your suggestions! We have addressed these semicolon and comma issues in the amended version of the manuscript.

Comment: The authors should correct the presentation style in lines 143, 203, 365: there is a period separated from the title number: for example, in section 3 it appears as follows: 3            .Results

Response: Thank you for picking up the formatting errors in these headlines. These errors have been corrected in the amended manuscript.

Comment: Please correct 

Congratulations to the authors

Response: Thank you!

Reviewer 3 Report

This paper aims to develop a conceptual framework for demand and supply of food credence attributes, by identifying the key actors, the actors' motivation to participate in the market, and the issue of information asymmetry. The studied topic is an interesting and promising one, that might be an important issue in food sector. However, I suggest the below corrections before the paper is published:

- The main motivation of the paper is not clear. Please highlight it correctly.

- Write at least 2 research questions.

- The search strings should be given clearly that are used in the literature survey. How are those key-words combined (e.g. by and/or) in the search process?

- Exclusion and inclusion process in the found paper should be explained.

- Any descriptive analysis on the found papers? 

- Figure 2 should be improved and extended by giving more findings in terms of framework. 

- Conclusion part seems very long. I suggest writing a separate conclusion part just summarising the paper along with the findings.

- Limitations of the paper as well as some future work suggestions should be given in the conclusion part.

Author Response

Thank you very much for your review. Please find below our responses to the comments. The revised manuscript is attached.

Comment: This paper aims to develop a conceptual framework for demand and supply of food credence attributes, by identifying the key actors, the actors' motivation to participate in the market, and the issue of information asymmetry. The studied topic is an interesting and promising one, that might be an important issue in food sector. However, I suggest the below corrections before the paper is published:

- The main motivation of the paper is not clear. Please highlight it correctly.

Response: The key motivation of this study is to fill the research gap which we identify in line 86-90: “However, missing in the literature is a clear overview that synthesizes a) the motivation for key stakeholders to participate in the market for food credence attributes, b) the identity of food credence attributes that key stakeholders provide, and c) current mechanisms to address the issue of information asymmetry among the stakeholders in the food system.”. In line 91 we state that this study aimed to address this research gap.

Comment: - Write at least 2 research questions.

Response: We have included the research questions of this study in lines 94-96: “Specific research questions included: What are the components of this conceptual framework? How do these components link with each other? Can information asymmetry be completely resolved using currently available mechanisms?”

Comment: - The search strings should be given clearly that are used in the literature survey. How are those key-words combined (e.g. by and/or) in the search process?

Response: We clarified that in lines 123-215 as “These terms were combined by using the AND (e.g., for food, credence, attribute) or OR (e.g., for assurance, certifying, verification) Boolean operator.”

Comment: - Exclusion and inclusion process in the found paper should be explained.

123, records were reviewed and publications were categorised  

Response: The exclusion process is outlined in line 127-132 as: “The inclusion process was based on an initial assessment of publication titles and abstracts that related to the research topic. Duplications records of low quality (e.g., some reports, topic related publications with limit research content value) were excluded in the initial assessment stage of publications (e.g., 127 records). A total of 243 publications were identified for inclusion in the qualitative analysis.” Figure 1 was updated by adding the numbers of records included and excluded in the analysis.

Comment: - Any descriptive analysis on the found papers? 

Response: We have included more detail in lines 130-147 about how the analysis was undertaken, e.g., theme identification and refinement, identification of links among themes. The identified themes and sub-themes are described in the results sections.

Comment: - Figure 2 should be improved and extended by giving more findings in terms of framework. 

Response: Figure 2 represents a visual framework model which was developed to identified themes and theme integration. As suggested, the figure now includes some additional information. The notes for Figure 2 offer additional information. Please also refer to the text in sections 3.1 and 3.2 which provides detailed information about the components of the visual framework model. Should further improvement/extension for Figure 2 be needed, please advise on the specific elements that should be included.

Comment: - Conclusion part seems very long. I suggest writing a separate conclusion part just summarising the paper along with the findings.

Response: The revised manuscript now includes a separate conclusion section (see lines 674-698) including the suggested paragraphs on limitations and further research needs.    

Comment: - Limitations of the paper as well as some future work suggestions should be given in the conclusion part.

Response: The limitations of this study are included the conclusion (see line 693-698). 

Reviewer 4 Report

The logic is not clear, the language is not standardized, and much of what needs to be explained does not go far enough.

1. poor logic and language in the abstract makes it difficult to get people interested in reading it.

2. The title of this work is a study of the demand and supply of food credit attributes. After reading the introduction, the current version lacks several dimensions: (1) This study focuses on food credit attributes, what are food credit attributes? Without a proper understanding of food credit attributes, it is difficult to interest the reader; (2) The presentation of the structure of this study is too normal and can be applied to other studies. (4) What are your contributions including theoretical contributions and managerial contributions? It is difficult to interest the reader without stating the contributions.

3. The introductory part of the literature review is missing.

4. After reading the literature review section, it is not clear to me what the research gap of this study is.

5. The research methodology is not clear. Is it just to present a framework? Further clarification is needed.

6. The literature review is more about providing your own perspective and not simply pasting and copying.

Author Response

Thank you very much for your review. Please find below our responses to the comments. The revised manuscript is attached.

Comments: The logic is not clear, the language is not standardized, and much of what needs to be explained does not go far enough.

Response: The revised manuscript underwent a spelling and grammar check. Please note that we acknowledge the complexity of this framework and that some elements of it can only be outlined in brief in this study due to word limits. We refer the interested reader to the cited literature for more details on these elements (see lines 156-159)).

Comment: 1. poor logic and language in the abstract makes it difficult to get people interested in reading it.

Response: The abstract has been revised by changing its structure and by briefly introducing the term food credence attributes (see lines 15-21). Furthermore, we added additional information about the findings (e.g., stakeholder groups with influential roles) as suggested by another reviewer. Hopefully, these changes contribute to raising readers’ interest in this study.

Comment: 2. The title of this work is a study of the demand and supply of food credit attributes. After reading the introduction, the current version lacks several dimensions: (1) This study focuses on food credit attributes, what are food credit attributes? Without a proper understanding of food credit attributes, it is difficult to interest the reader; (2) The presentation of the structure of this study is too normal and can be applied to other studies. (4) What are your contributions including theoretical contributions and managerial contributions? It is difficult to interest the reader without stating the contributions.

Response: (1) The title of the manuscript has been changed to “Food credence attributes: A conceptual framework of market stakeholders, their motives, and mechanisms to address information asymmetry”. The term food credence attribute is now briefly introduced in the abstract (see lines 14-21). More detailed definitions of different food attributes, including search, experience and credence attributes and the associated issue of information asymmetry remain unchanged in the introduction (lines 55-65). (2) The structure of the manuscript follows the guidelines of the journal. Please clarify should your comment refer to any other parts of the manuscript. (4) The findings from this study offer a theoretical contribution which may improve food supply chain stakeholders’ (e.g., supply chain actors, scientists, policy makers) understanding about the components of the credence food system and their integration as well as the drivers for change in this system. This has been added to the abstract (lines 35-38) and the introduction (lines 101-104).

Comment: 3. The introductory part of the literature review is missing.

Response: It is entirely clear what the reviewer means by this comment. We provide an introduction and a methods section which outline the background to this study and how the literature review was conducted. The results section (see lines 152-173) offers an introduction about the findings of the literature review (e.g., the framework and two key thematic parts which are briefly introduced before providing more details in the subsections. The authors would be grateful for clarifications from the reviewer on which specific parts of the manuscript (literature review?) may require further introduction.

Comment: 4. After reading the literature review section, it is not clear to me what the research gap of this study is.

The research gap which this study is aiming to address is stated in lines 86-90: “However, missing in the literature is a framework which synthesizes a) the motivation for key stakeholders to participate in the market for food credence attributes, b) the identity of food credence attributes that key stakeholders provide, and c) current mechanisms to address the issue of information asymmetry among the stakeholders in the food system.” The synthesis of these aspects of the credence food system are expected to offer a theoretical contribution which may improve food supply chain stakeholders’ (e.g., supply chain actors, scientists, policy makers) understanding about the components of the credence food system and their integration as well as the drivers for change in this system (see lines 101-104).

Comments: 5. The research methodology is not clear. Is it just to present a framework? Further clarification is needed.

Response: The revised manuscript includes additional information about the methodology used in this study (see lines 117-147), including Boolean operators used for publication search, exclusion and inclusion criteria, number of publications considered, process of theme and link identification.

Comment 6: The literature review is more about providing your own perspective and not simply pasting and copying.

In the results section we summaries/synthesize the findings from the literature review. Throughout the results section we cite relevant publications which support the inclusion of the framework’s components and linkages between them. The framework and narrative represent how the authors summarize and interpret the literature to achieve the aim of this study. This approach is commonly used in integrative literature reviews (e.g., Torraco 2005; Torraco 2016; Wu et al. 2021). The authors of the manuscript confirm that they have not copied and pasted sections from other publications. The journal editor may have used available software to confirm that the presented manuscript is entirely the work of the authors.

References

Torraco, R.J. (2005). Writing Integrative Literature Reviews: Guidelines and Examples, 4, 356-367. doi: https://doi.org/10.1177/1534484305278283

Torraco, R.J. (2016). Writing Integrative Literature Reviews: Using the Past and Present to Explore the Future, 15, 404-428. doi: https://doi.org/10.1177/1534484316671606

Wu, W., Zhang, A., van Klinken, R.D., Schrobback, P. and Muller, J.M. (2021). Consumer Trust in Food and the Food System: A Critical Review, 10, 2490. doi: https://doi.org/10.3390/foods10102490

Round 2

Reviewer 1 Report

Thank you for your attention to revise the article that will be easy to understand for reader.

Reviewer 3 Report

The paper is revised thoroughly. Hence, I suggest it is accepted to be published.

Reviewer 4 Report

Agree to accept. Good luck.